# Efficient-DLM: From Autoregressive to Diffusion Language Models, and Beyond in Speed

Yonggan Fu [* 1]  Lexington Whalen [† * 2]  Zhifan Ye [† 2]  Xin Dong [1]  Shizhe Diao [1]  Jingyu Liu [† 3]  Chengyue Wu [† 4]
Hao Zhang [1]  Enze Xie [1]  Song Han [1 5]  Maksim Khadkevich [1]  Jan Kautz [1]  Yingyan (Celine) Lin [1 2]
Pavlo Molchanov [1]

## Abstract

Diffusion language models (dLMs) have emerged as a promising paradigm enabling parallel generation, but their learning efficiency lags behind that of autoregressive (AR) language models when trained from scratch. To this end, we study AR-to-dLM conversion, which transforms pretrained AR models into efficient dLMs that excel in speed while preserving AR models' task accuracy. We achieve this by identifying limitations in the attention patterns and objectives of existing AR-to-dLM methods and then proposing methodologies and actionable insights for scalable AR-to-dLM conversion. Specifically, we first systematically compare different attention patterns and find that maintaining pretrained AR weight distributions is key to effective AR-to-dLM conversion. Accordingly, we introduce a continuous pretraining scheme with a block-wise attention pattern. We find that, in addition to block-wise attention's known benefit of enabling KV caching, its block-wise causality better preserves pretrained AR models' weight distributions, leading to a win–win in accuracy and efficiency. Second, to mitigate the training–test gap in mask token distributions (uniform vs. highly left-to-right), we propose a position-dependent token masking strategy that assigns higher masking probabilities to later tokens during training to better mimic test-time behavior. These studies lead to the Efficient-DLM model family, which outperforms state-of-the-art AR models and dLMs in accuracy–throughput trade-offs; for example, our

Efficient-DLM-8B achieves +5.4%/+2.7% higher accuracy with 4.7×/2.8× higher throughput compared to Dream-7B and Qwen3-4B, respectively.

**Models on Hugging Face:** Efficient-DLM Model Family

## 1. Introduction

Diffusion language models (dLMs) (He et al., 2022; Sahoo et al., 2024; Nie et al., 2025; Ye et al., 2025) have recently emerged as an alternative paradigm to autoregressive (AR) modeling for large language models (LLMs), driven by their promise of higher throughput via parallel, non-autoregressive generation. However, successful scaling of dLMs to larger model sizes has been restricted by prohibitive training costs (Nie et al., 2024). This is because AR models learn only left-to-right modeling, while dLMs learn all possible permutations (Xue et al., 2025), which is more difficult and requires longer training.

This work leverages pretrained AR models for initialization and systematically explores how to continuously pretrain them into dLMs that achieve high throughput while preserving task accuracy. We achieve this by identifying limitations in the attention patterns and objectives of existing AR-to-dLM methods and proposing a continuous pretraining scheme that features a block-wise attention pattern (Arriola et al., 2025) and position-dependent token masking.

Specifically, our extensive study of different attention patterns shows that a block-wise attention pattern, beyond its known benefit of enabling KV caching, better preserves the weight distributions of pretrained AR models than the fully bidirectional training adopted in prior work (Nie et al., 2025; Ye et al., 2025), which is key to effective AR-to-dLM conversion. These findings provide a practical guideline for the community: block-wise attention (with each block conditioned on clean context during training) is a preferred choice for AR-to-dLM conversion, achieving a win–win in accuracy and efficiency. Second, we identify a mismatch between training-time uniform token masking and test-time confidence-based token sampling. To bridge this gap and improve downstream accuracy, we propose a

---

[*]Co-first author [†]Work done during internships at NVIDIA [1]NVIDIA [2]Georgia Institute of Technology [3]University of Chicago [4]University of Hong Kong [5]MIT. Correspondence to: Yonggan Fu <yongganf@nvidia.com>, Pavlo Molchanov <pmolchanov@nvidia.com>.

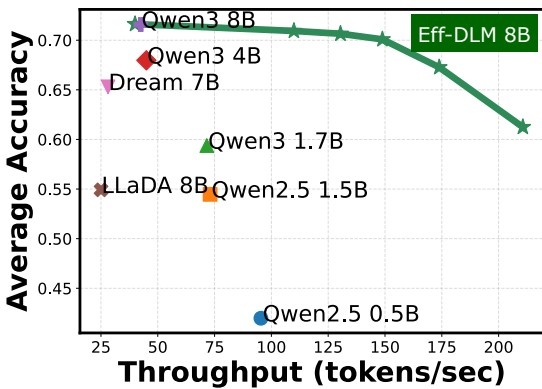

*Figure 1.* The accuracy–throughput trade-offs between Efficient-DLM-8B and SOTA AR/dLMs, averaged over 12 tasks.

**Takeaways for effective AR-to-dLM conversion**

- *Attention pattern is key to AR-to-dLM conversion*: Block-wise attention better preserves pretrained AR weight distributions than fully bidirectional modeling; Conditioning each corrupted block on clean context is essential for model accuracy.

- *Training block size matters*: Too-small block sizes lack sufficient context for denoising, while too-large block sizes induce excessive corruption.

- *Evaluation block size matters*: Training with proper block sizes can generalize well to other evaluation block sizes; Larger evaluation block sizes generally provide more opportunities for parallel decoding.

- *Whether to preserve the token shift of AR models*: We find it unnecessary and potentially harmful.

- *Left-to-right generation tendency*: dLMs exhibit this tendency during parallel generation, and mimicking it in training can boost generation quality.

- *Training dynamics*: Likelihood estimation improves steadily with training, allowing for more aggressive parallel decoding.

position-dependent token masking strategy. This approach builds on the observation that dLMs retain a left-to-right generation tendency due to the autoregressive nature of language, and thus incorporates the prior that, as the input becomes less corrupted (i.e., closer to complete denoising), more tokens should be masked toward the end of each block.

Furthermore, under this training framework, we analyze attention patterns, training dynamics, and other design choices for scalable AR-to-dLM conversion, and introduce the Efficient-DLM model family, which outperforms both AR and dLM baselines with improved accuracy–throughput trade-offs. For example, as shown in Fig. 1, our Efficient-DLM-8B maintains accuracy comparable to Qwen3-8B and achieves +5.4%/+2.7% higher accuracy with 4.7×/2.8× higher throughput compared to Dream-7B and Qwen3-4B, respectively.

We expect these findings to provide practical guidelines for realizing dLMs' promise of faster, more efficient generation, and to inspire new dLM paradigms. The key takeaways and insights from this work are summarized on the right.

## 2. Efficient-DLM: Attention Pattern Analysis

### 2.1. Analyzing Different Attention Patterns

**Fully bidirectional attention in existing works.** Existing works that transform AR models into dLMs (Gong et al., 2025a; Ye et al., 2025) adopt fully bidirectional modeling, i.e., the entire sequence is randomly corrupted and all tokens are visible to each other, as shown in Fig. 2 (a) and (b). This training scheme suffers from the following drawbacks: (1) fully bidirectional attention increases the difficulty of applying KV caching; (2) the context is overly corrupted, particularly for later tokens, which increases training difficulty; (3) the fully bidirectional attention pattern diverges from the causality of the AR initialization, resulting in larger weight drifts from pretrained AR models.

**Block-wise attention.** In light of these limitations, an enhanced attention pattern is the block-wise attention shown in Fig. 2 (c), which remains causal across blocks and adopts bidirectional modeling within each block. This design enables the use of KV caching at test time, and its block-wise causality is closer to the token-wise causality of pretrained AR models, potentially better preserving their abilities.

**Block-wise attention with each block conditioned on clean context.** The drawback of the block-wise attention in Fig. 2 (c) is that it may cause a training–test gap: when performing block-wise decoding at test time, the context preceding a noisy block has already been decoded without mask tokens; however, the attention pattern in Fig. 2 (c) cannot ensure this during training, as the context of each block still contains mask tokens. To resolve this training–test gap, our work employs block-wise attention with each block conditioned on clean context, as shown in Fig. 2 (d), following (Arriola et al., 2025).

This design ensures that each block is conditioned only on clean context during training, mimicking the block-wise decoding process at test time, where all previous blocks are fully completed without mask tokens. This is achieved by concatenating the noisy tokens and the clean tokens as dLM inputs and applying the special attention mask shown in Fig. 2 (d). Such an attention pattern allows for seamless use of the KV cache for improved efficiency, constrains corrup-

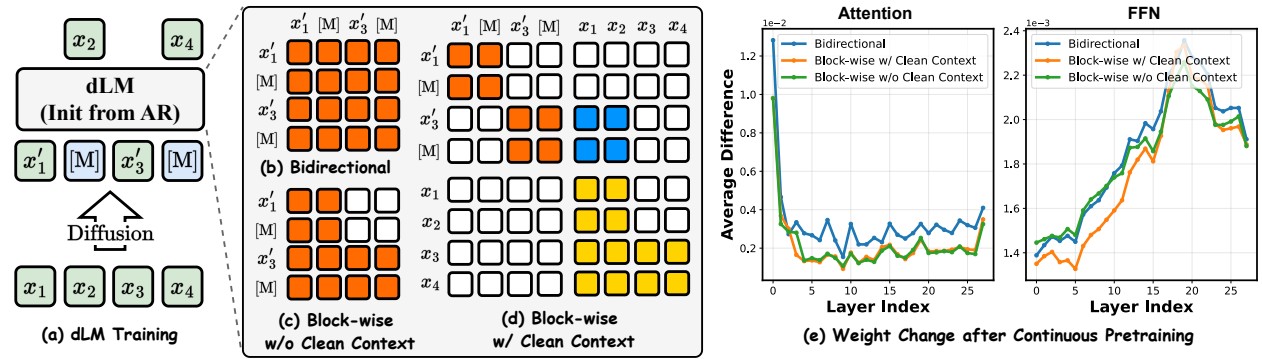

*Figure 2.* Visualizing continuous pretraining of dLMs with different attention patterns from pretrained AR models. (b) and (c) show bidirectional attention and block-wise attention without clean context, respectively, using a block size of 2 as an example. (d) illustrates the block-wise attention with clean context (using a block size of 2 as an example), where ■ denotes attention among noisy tokens, ■ denotes attention from noisy tokens to clean-context tokens, and ■ denotes attention within the clean context. (e) shows weight changes in the attention and feed-forward network (FFN) layers after training under these three attention patterns.

*Table 1.* Comparison of dLM training schemes on Qwen2.5-1.5B. Rows (a)/(b) correspond to the original Qwen2.5-1.5B and Dream's training scheme (Ye et al., 2025), respectively. Row (g) achieves the best performance (block-wise attention, clean context, no token shift).

| Row ID | Attn Pattern | Clean Context | Token Shift | KV Cache | Human -Eval | Human -Eval Plus | MBPP | MBPP Plus | GSM8K | Minerva Math | Avg |
|---|---|---|---|---|---|---|---|---|---|---|---|
| a | AR | - | ✔ | ✔ | 36.59 | 29.88 | 43.6 | 59.52 | 54.74 | 26.40 | 41.79 |
| b | Bidirectional | - | ✔ | ✗ | 15.85 | 12.20 | 16.2 | 24.34 | 28.96 | 11.08 | 18.10 |
| c | Bidirectional | - | ✗ | ✗ | 19.51 | 15.24 | 17.2 | 24.34 | 28.20 | 11.22 | 19.29 |
| d | Block-wise | ✗ | ✔ | ✔ | 31.10 | 25.61 | 23.6 | 36.77 | 38.44 | 13.88 | 28.23 |
| e | Block-wise (2×) | ✗ | ✔ | ✔ | 26.22 | 22.56 | 26.0 | 42.33 | 36.69 | 12.56 | 27.73 |
| f | Block-wise | ✔ | ✔ | ✔ | 38.41 | 33.54 | 33.0 | 48.68 | 51.48 | 21.04 | 37.69 |
| g | Block-wise | ✔ | ✗ | ✔ | 39.02 | 34.76 | 34.0 | 48.15 | 52.99 | 21.56 | 38.41 |

tion within each block to maintain a cleaner context, and preserves block-wise causality, thereby mitigating weight drift and better inheriting the capabilities of the AR model.

More formally, let $\mathbf{x} = (x_1, \ldots, x_L)$ be a sequence partitioned into $B$ contiguous blocks $\mathbf{x}^b$ of length $L' = L/B$, and let $q(\tilde{\mathbf{x}}_t^b \mid \mathbf{x}^b)$ denote the corruption process that produces the noisy input $\tilde{\mathbf{x}}_t^b$ at noise level $t \in (0, 1]$. Our training objective is as follows:

$$\mathcal{L}(\theta) = \mathbb{E}_{t \sim \mathcal{U}[0,1]} \, \mathbb{E}_{\tilde{\mathbf{x}}_t^b \sim q(\cdot|\mathbf{x}^b)} \left[ -\frac{1}{t} \sum_{b=1}^{B} \log p_\theta(\mathbf{x}^b \mid \tilde{\mathbf{x}}_t^b, \mathbf{x}^{<b}) \right]$$
(1)

where $p_\theta(\mathbf{x}^b \mid \tilde{\mathbf{x}}_t^b, \mathbf{x}^{<b})$ denotes the denoising of the $b$-th block based on the corrupted input $\tilde{\mathbf{x}}_t^b$ and the clean context $\mathbf{x}^{<b}$. Similarly, block-wise attention without using clean context in Fig. 2 (c) can be formulated by replacing the clean context $\mathbf{x}^{<b}$ in Eq. 1 with the corrupted context $\tilde{\mathbf{x}}^{<b}$.

Unlike prior block diffusion methods trained from scratch (Arriola et al., 2025), we initialize $\theta$ from a pretrained AR model that is trained with the autoregressive loss $\mathcal{L}_{AR}(\theta) = -\sum_{\ell=1}^{L} \log p_\theta(x_\ell \mid x_{<\ell})$, and then adapt it through continuous pretraining using the loss in Eq. 1. This initialization allows for rapid AR-to-dLM conversion, which requires (1) adapting weights to new attention pat-

terns and (2) avoiding large weight drifts to better preserve the original model's ability.

## 2.2. Comparison of Different Attention Patterns

We study the impact of three key design factors on attention patterns in AR-to-dLM conversion: (1) fully bidirectional vs. block-wise; (2) whether to keep clean context: using clean context $\mathbf{x}^{<b}$ or corrupted context $\tilde{\mathbf{x}}^{<b}$ in Eq. 1; and (3) whether to perform token shift, i.e., predicting the next token as in AR models or directly predicting the mask tokens themselves. Previous works (Gong et al., 2025a; Ye et al., 2025) find that preserving token shift when initializing from AR models is beneficial, and we revisit this design choice under more advanced training schemes.

**Settings.** We adopt Qwen2.5-1.5B (Team, 2024) as the AR initialization and perform continuous pretraining for 50B tokens on a mixed dataset comprising (Nano, 2025; Zhou et al., 2025; Fujii et al., 2025). For block-wise training, we adopt a block size of 16, and provide further analysis on block sizes in Sec. 2.3. The initial learning rate is set to 1e-5 and decayed to 3e-6 using a cosine schedule with the AdamW optimizer. An analysis of the learning rate

is provided in Appendix D. We evaluate downstream task accuracy on six generation tasks, including HumanEval, HumanEval Plus, MBPP, MBPP Plus, GSM8K, and Minerva Math, using lm-evaluation-harness (Gao et al., 2024).

**Importance of block-wise attention.** As shown in Tab. 1, compared to bidirectional attention in Row (c), block-wise attention (even without clean context) in Row (d) can boost average accuracy by 8.94%. This implies that block-wise attention better preserves block-wise causality and thus maintains the pretrained AR model's abilities more effectively than bidirectional attention, in addition to the benefit of native KV caching. Furthermore, visualizations of weight changes after continuous pretraining in Fig. 2 (e) show that bidirectional attention leads to larger weight drifts from pretrained weights in both the attention and FFN layers, ultimately causing larger accuracy drops.

**The impact of clean context.** Based on the comparison between Rows (d) and (f) in Tab. 1, conditioning each block on clean context during training is critical, yielding a 9.46% accuracy improvement over using noisy context, where both adopt block-wise attention. We further train the noisy-context case with a doubled token budget, i.e., extend Row (d) to Row (e), to account for the increased sequence length caused by concatenating noisy and clean tokens in the setting of Row (f). However, comparing Rows (e) and (f) in Tab. 1 shows that doubling training tokens on corrupted context cannot effectively recover the accuracy, whereas training on fewer tokens with clean context yields substantially higher accuracy.

In addition, Fig. 2 (e) shows that training with block-wise attention without clean context leads to larger weight drifts in FFN layers compared to training with clean context.

**Whether to perform token shift.** We find that token shift is unnecessary, and its removal consistently improves accuracy across settings, as evidenced by the comparison between Rows (b) & (c) and Rows (f) & (g) in Tab. 1. This indicates that (1) the token shift inherent to AR models can be easily adapted into the no-token-shift setting, and (2) predicting the mask token itself (without token shift) is easier than predicting the next token of a masked position. We hypothesize that the latter is harder because the model must handle two tasks simultaneously: inferring the mask token and predicting the following token.

Based on the above experiments, we draw the following takeaways and adopt the identified best-performing scheme as the default in the remainder of this study.

> **Takeaways:** When continuously pretraining from an AR model, a block-wise attention pattern with clean context and without token shift emerges as a promising training scheme to deliver dLMs.

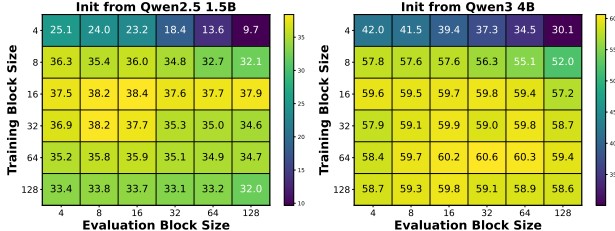

*Figure 3.* The average accuracy achieved by different training–evaluation block size pairs.

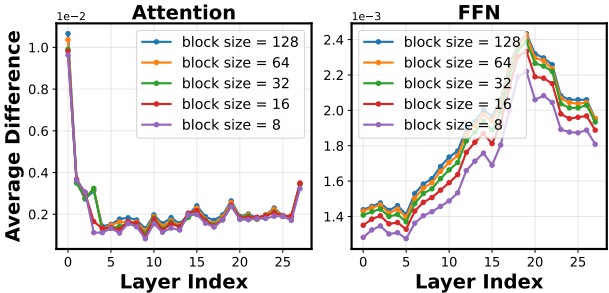

*Figure 4.* The weight changes in attention and FFN layers after training with different block sizes.

### 2.3. Analysis of the Optimal Block Sizes

The next question is the optimal block size for training and evaluation. Intuitively, larger context sizes provide richer context with more visible future tokens, but at the same time introduce more corruption, i.e., the last tokens in a block encounter noisier past context. This makes it critical to select a proper block size that balances both aspects.

**Settings.** We perform continuous pretraining on top of Qwen2.5-1.5B (Team, 2024) and Qwen3-4B (Yang et al., 2025) for 50B and 25B tokens, respectively, using different training block sizes. Other configurations are the same as in Sec. 2.2. We then evaluate each trained model with different evaluation block sizes and report the average accuracy across six generation tasks (HumanEval, HumanEval Plus, MBPP, MBPP Plus, GSM8K, and Minerva Math) for each training–evaluation block size pair in Fig. 3.

**Observations on training block sizes.** As shown in Fig. 3, we observe that (1) for both model scales, too-small training block sizes generally lead to suboptimal accuracy, potentially because the context is not sufficiently rich to predict the corruptions. (2) Larger-scale models are more tolerant of larger training block sizes, which introduce more corruptions but also provide richer context. In contrast, small-scale models have a smaller sweet-spot training block size, beyond which the more corrupted context leads to degraded accuracy. (3) Training with an appropriate block size can transfer well to other evaluation block sizes. This can be inferred from the attention map in Fig. 2 (d), where the model trained with a single block size can see varying numbers of tokens participating in the attention mechanism. This differs

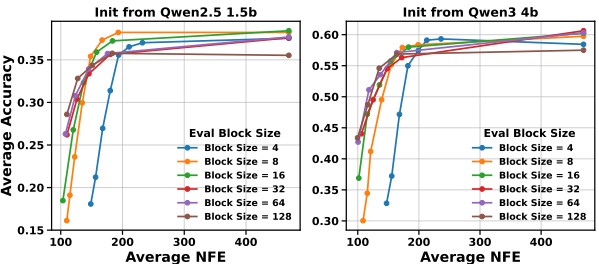

*Figure 5.* The acc-NFE trade-offs of different eval block sizes.

from the fully bidirectional attention in Fig. 2 (b), where the model always sees the same number of tokens participating in attention, necessitating additional techniques such as random sequence length truncation (Nie et al., 2025) to generalize to other sequence lengths.

We also visualize the weight changes after continuous pre-training with different block sizes on Qwen2.5-1.5B in Fig. 4. In general, larger training block sizes lead to larger weight changes. This indicates a trade-off between maintaining original abilities and adapting to new attention patterns, and the sweet-spot block size balances both aspects.

> **Takeaways:** There exists a sweet-spot training block size: too-small block sizes lack sufficient context, while too-large block sizes induce excessive corruption and weight changes.

**Observations on evaluation block sizes.** We evaluate trained dLMs with different block sizes and adopt confidence-based sampling (Wu et al., 2025) with different confidence thresholds to control the number of function evaluations (NFEs) (Ou et al., 2024). The lower the NFE, the more tokens are generated in parallel. As shown in Fig. 5, we observe that (1) larger evaluation block sizes generally lead to higher accuracy when performing more aggressive token generation with lower NFEs, potentially because larger block sizes provide greater opportunities for parallel token generation; and (2) with larger NFEs, there is no clearly optimal evaluation block size, and moderate block sizes generally yield comparable results.

> **Takeaways:** Although a proper training block size can generalize to other evaluation block sizes, larger evaluation block sizes favor more aggressive parallel token generation.

## 3. Position-dependent Token Masking

### 3.1. The Training-Test Gap in Token Masking

Existing dLMs (Nie et al., 2025; Ye et al., 2025; Arriola et al., 2025) typically adopt uniform token masking, where mask tokens are randomly sampled from a uniform distribu-

tion based only on the noise level $t$, independent of token positions. However, we find that at inference time, when performing confidence-based sampling (Nie et al., 2025; Ye et al., 2025), the denoised tokens are not uniformly distributed; instead, they show a clear left-to-right tendency.

To demonstrate this, we visualize the average number of denoising steps required at each token position in a block on the GSM8K dataset using the trained diffusion Qwen2.5-1.5B model from Sec. 2.2 in Fig. 6 (a). Specifically, we visualize two cases, with and without parallel token generation, using a confidence threshold (Wu et al., 2025). We observe that the average number of denoising steps increases with the positions in a block, exhibiting a notable left-to-right tendency due to the autoregressive nature of language. In addition, as a concrete example, we also show the confidence distribution within a block across different denoising steps for one example from GSM8K in Fig. 6 (b), with red boxes marking tokens that are decoded and finalized. We can see that tokens tend to have higher confidence scores once their neighboring tokens have been decoded and are generally decoded from left to right. In other words, as the denoising process approaches completion, mask tokens are more likely to appear near the block end. As such, uniform token masking during training and confidence-based sampling during inference create a training–test gap.

In addition, we also visualize the average loss at each token position within a block in Fig. 6 (c), averaged over all blocks for 200 samples. We observe that the later mask tokens in a block are generally harder cases with larger losses due to more corrupted context, potentially requiring more learning. This also indicates that a more strategic token masking scheme that also considers token position is desirable.

### 3.2. Our Token Masking Strategy

To demonstrate the importance of mitigating the training–test gap, we introduce the concept of position-dependent token masking. Specifically, for a sequence $\mathbf{x} = (x_1, \ldots, x_L)$ and a given noise level $t$, conditioned on $t$ and the token position $i \in [L']$ within one block, the masking probability of each token position is set as

$$w_i(t) = \exp(\beta(1-t)i), \tag{2}$$

where $\beta \geq 0$ is a hyperparameter controlling the strength of the positional bias. Specifically, $\beta = 0$ leads to uniform sampling, and larger $\beta$ indicates a stronger positional bias. The set of mask tokens is drawn from this distribution by normalizing the weights and then performing Gumbel-top-$k$ sampling (Huijben et al., 2022), where $k = \lfloor tL' \rfloor$ is the per-block mask token count. When $t \to 0$, corresponding to the end of denoising, $w_i(t)$ assigns larger weights to later tokens, i.e., mask tokens are more likely to appear near the block end, echoing the test-time pattern in Sec. 3.1. When $t \to 1$, corresponding to noisier inputs, $w_i(t)$ becomes more

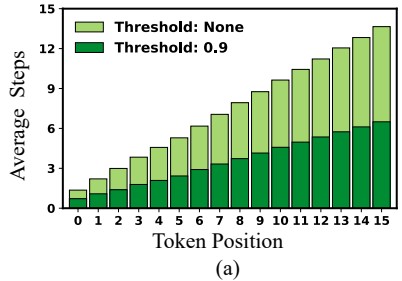
(a)

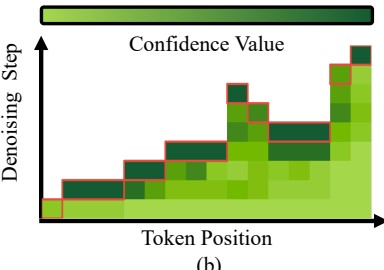
(b)

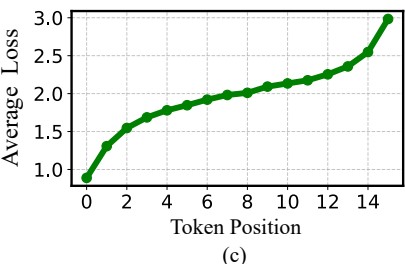
(c)

*Figure 6.* (a) The average number of denoising steps required at each token position on GSM8K using diffusion Qwen2.5-1.5B with two different confidence thresholds, where "None" denotes one token per step. (b) The confidence distribution across different denoising steps for an example from GSM8K, with red boxes marking tokens that are decoded. (c) The average loss at each token position within a block.

| Setting | TPF=1 | TPF=2.8 | TPF=4 | TPF=5.6 | Avg Diff |
|---------|-------|---------|-------|---------|----------|
| uniform | 60.27 | 57.12 | 51.11 | 33.99 | - |
| right-to-left | 38.21 | 27.55 | 18.60 | 14.13 | - |
| $\lambda$=0.25 | 60.41 (+0.14) | 58.56 (+1.44) | 50.97 (-0.14) | 34.55 (+0.56) | +0.50 |
| $\lambda$=0.1 | 62.02 (+1.75) | 58.79 (+1.67) | 53.75 (+2.64) | 38.37 (+4.38) | +2.61 |
| $\lambda$=0.05 | 60.51 (+0.24) | 57.68 (+0.56) | 53.06 (+1.95) | 37.38 (+3.39) | +1.54 |

*Table 2.* Comparing token masking schemes based on the average accuracy across six generation tasks under varying parallel decoding settings, measured in tokens per forward (TPF).

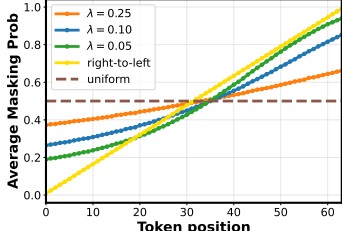

*Figure 7.* The average masking probability of each token position.

uniform, i.e., mask tokens are sampled more uniformly. The increased masking probability for later tokens also involves more hard cases with larger losses. We also note that related work (Wu et al., 2023) leverages the AR nature of language and designs a left-to-right noise schedule for continuous dLMs, whereas our work focuses on discrete dLMs.

### 3.3. Comparison of Token Masking Schemes

**Settings.** We apply position-dependent token masking with different $\beta$ to the training of diffusion Qwen3-4B with block size 64 on 25B tokens. In practice, instead of directly setting $\beta$, we parameterize the positional prior using a half-life ratio $\lambda = \ln 2/(\beta L') \in (0, 1]$. The half-life ratio $\lambda$ is the fraction of a block length over which, under maximal tilt $t \to 0$, the positional weight changes by a factor of two. Thus, the lower the value of $\lambda$, the stronger the positional prior.

We compare position-dependent token masking with different $\lambda$ values against uniform token masking (i.e., $\lambda \to \infty$) and right-to-left masking (i.e., $\lambda \to 0$), which always masks the rightmost $k$ tokens. The average masking probability of each position within a block throughout training is shown in Fig. 7. The average accuracy on the six generation tasks in Sec. 2.3, under different parallel decoding settings in terms of tokens per forward (TPF), i.e., the number of decoded tokens per denoising step using confidence-based sampling (Wu et al., 2025), is presented in Tab. 2.

**Observations.** As shown in Tab. 2, we observe that (1) progressively increasing positional priors with lower $\lambda$ leads to

improved average accuracy; (2) positional priors are particularly beneficial under more aggressive parallel decoding settings, with up to a 4.38% average accuracy improvement; and (3) Positional priors should not be blindly increased, as extreme right-to-left masking degrades performance, likely because the model is forced to predict consecutive tokens at block ends without leveraging bidirectional context.

> **Takeaways:** dLMs exhibit a left-to-right tendency during parallel generation, and mimicking this tendency in training can boost generation quality.

The key value here is to introduce this design factor, and we hope it can inspire more advanced token masking schemes.

## 4. Training Dynamics Analysis

dLM training with the objective in Eq. 1 improves the masked denoising likelihood, but how this improved likelihood estimation translates into downstream task accuracy and parallel token generation ability remains unclear. In this section, we study the training dynamics of dLMs by visualizing their task performance evolution.

**Settings.** We train Qwen2.5-1.5B for 200B tokens with the same setting as in Sec. 2.3, and evaluate on both generation and likelihood-based tasks, where accuracy is computed by estimating and selecting the largest likelihood among multiple choices (Nie et al., 2025). We visualize the accuracy evolution on likelihood tasks and the accuracy–NFE

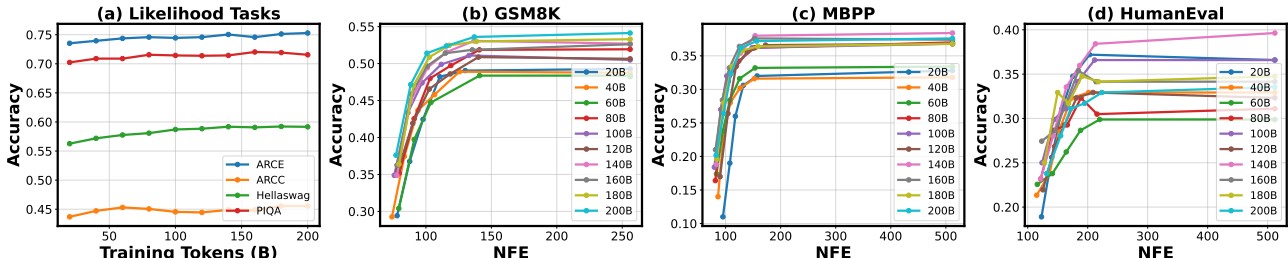

*Figure 8.* (a) The accuracy evolution on likelihood tasks. (b–d) The accuracy–NFE trade-offs under different training token budgets.

*Table 3.* Benchmarking against SOTA AR models and dLMs on 12 tasks spanning coding, math, factual knowledge, and commonsense reasoning (CR), reporting average accuracy per category. TPF denotes tokens per forward, and TPS refers to throughput measured on an NVIDIA H100 GPU with a batch size of 1. Detailed per-task accuracy is provided in Appendix B.

| Type | Model | TPF | TPS (tok/sec) | Coding | Math | MMLU | CR | Avg. |
|------|-------|-----|---------------|--------|------|------|-----|------|
| AR | Llama-3.2-1B | 1.00 | 143.91 | 24.45 | 4.98 | 30.98 | 60.62 | 36.82 |
| | Qwen2.5-0.5B | 1.00 | 99.93 | 31.90 | 25.97 | 47.65 | 55.31 | 41.98 |
| | Qwen2.5-1.5B | 1.00 | 73.03 | 42.17 | 46.98 | 60.96 | 66.00 | 54.47 |
| dLM (Ours) | Efficient-DLM-1.5B | 2.33 | 158.89 | 42.33 | 42.28 | 57.63 | 62.58 | 52.04 |
| | | 2.69 | 184.48 | 41.79 | 41.80 | 57.63 | 62.58 | 51.77 |
| AR | Qwen3-1.7B | 1.00 | 71.59 | 54.22 | 54.15 | 62.53 | 64.99 | 59.39 |
| | Qwen3-4B | 1.00 | 47.13 | 63.85 | 66.27 | 73.19 | 70.91 | 67.97 |
| | Qwen3-8B | 1.00 | 42.51 | 68.45 | 69.84 | 76.93 | 73.71 | 71.58 |
| dLM | LLaDA-8B | 1.00 | 25.04 | 38.10 | 49.13 | 65.86 | 68.50 | 54.92 |
| | Dream-7B | 1.00 | 28.11 | 58.92 | 58.39 | 67.00 | 72.83 | 65.30 |
| dLM (Ours) | Efficient-DLM-4B | 2.52 | 119.33 | 61.37 | 68.56 | 71.80 | 70.87 | 67.39 |
| | | 3.01 | 130.24 | 60.96 | 68.10 | 71.80 | 70.87 | 67.18 |
| dLM (Ours) | Efficient-DLM-8B | 2.74 | 109.78 | 65.64 | 68.52 | 77.22 | 74.88 | 70.93 |
| | | 3.27 | 130.71 | 64.95 | 68.21 | 77.22 | 74.88 | 70.65 |

trade-off across different training token budgets in Fig. 8.

**Observations and analysis.** We observe that (1) with relatively low training cost (on the order of 10B tokens), dLMs converted from pretrained AR models can largely recover task accuracy. (2) Longer training with more iterations consistently improves likelihood estimation and yields higher accuracy on likelihood-based tasks. The average accuracy on generation tasks, without considering parallel token generation (i.e., the rightmost points of each curve in Fig. 8 (b–d)), also improves, though with fluctuations on certain tasks. (3) Improved likelihood estimation allows for more aggressive parallel token generation, as reflected in the enhanced accuracy–NFE trade-off with longer training. This indicates that stronger likelihood estimation produces more accurate and reliable confidence scores, thereby improving generation quality under confidence-based sampling.

This also indicates that parallel token generation ability is another dimension for evaluating a dLM's performance:

dLMs with comparable accuracy when denoising one token per step can exhibit notable accuracy gaps when performing aggressive parallel token generation.

> **Takeaways:** dLMs' ability to perform more aggressive parallel generation improves with better likelihood estimation, which can be induced by longer training on more tokens.

## 5. Efficient-DLM: A New Family of dLMs

Combining previous insights, we develop the Efficient-DLM family with three sizes (1.5B/4B/8B), continuously pretrained from Qwen2.5-1.5B (block size 16) and Qwen3-4B/Qwen3-8B (block size 64), respectively. Our Efficient-DLM family integrates the identified best attention pattern in Sec. 2 and the position-dependent token masking with $\lambda = 0.1$ in Sec. 3. Motivated by the training dynam-

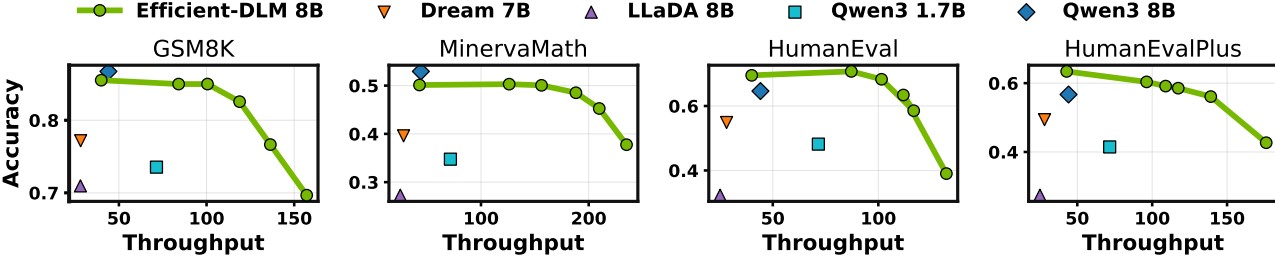

*Figure 9.* Visualizing the accuracy-throughput trade-off of different models across different generation tasks.

ics discussed in Sec. 4, we train for longer (300B tokens for Efficient-DLM-1.5B/4B and 500B tokens for Efficient-DLM-8B) using a mixed dataset comprising (Nano, 2025; Zhou et al., 2025; Fujii et al., 2025), adopting the AdamW optimizer and an initial learning rate of 1e-5 with cosine decay. All models are trained on 128 NVIDIA H100 GPUs.

### 5.1. Benchmark with SOTA AR and dLMs

We benchmark our Efficient-DLM against SOTA AR LMs (Qwen3 (Yang et al., 2025), Qwen2.5 (Team, 2024), Llama-3.2 (Grattafiori et al., 2024)) and dLMs (LLaDA (Nie et al., 2025) and Dream (Ye et al., 2025)) in Tab. 3. The benchmark covers 12 tasks, including math (GSM8K, Minerva Math), coding (HumanEval, HumanEval Plus, MBPP, MBPP Plus), factual knowledge (MMLU), and commonsense reasoning (ARC-C, ARC-E, Hellaswag, PIQA, Winogrande), as well as throughput measured on an NVIDIA H100 GPU with a batch size of 1. For each instance of Efficient-DLM, we report results under different parallel decoding settings with varying confidence thresholds (Wu et al., 2025). More details are provided in Appendix A.

**Observations.** As shown in Tab. 3, we observe that (1) compared to SOTA dLMs, our Efficient-DLM achieves both higher accuracy and efficiency. For example, Efficient-DLM-8B delivers 5.35% higher average accuracy with 4.65× throughput over Dream-7B, benefiting from the block-wise attention design over fully bidirectional modeling (see Sec. 2.2). (2) Compared to SOTA AR LMs, our Efficient-DLM attains better accuracy–throughput trade-offs. For instance, Efficient-DLM-8B/4B achieves 2.77×/1.82× throughput with +2.68%/+7.79% accuracy over Qwen3-4B/1.7B, respectively.

### 5.2. One-for-All: Flexible Acc–Efficiency Trade-offs

Beyond efficiency, another advantage of dLMs is their one-for-all flexibility: a single dLM can balance accuracy and throughput to suit different deployment scenarios. This is achieved by controlling parallel token generation via different confidence thresholds (Wu et al., 2025). Fig. 9 shows the one-for-all flexibility of our Efficient-DLM-8B across four math and coding tasks. We observe that a single Efficient-

DLM-8B achieves better accuracy–throughput frontiers than the AR Qwen3 family from 1.7B to 8B, demonstrating its promise for one-for-all deployment. Throughput results under large batch sizes are provided in Appendix C.

### 5.3. Advantages of dLMs in Text Embedding

We further highlight that, thanks to their ability for bidirectional modeling, dLMs are more promising than AR models for tasks requiring high-quality text embeddings. To demonstrate this, we evaluate our Efficient-DLM against AR Qwen models on text embedding tasks, benchmarking 15 datasets from the MTEB benchmark (Muennighoff et al., 2022) across six categories, following the ablation setup from LLM2Vec (BehnamGhader et al., 2024). As shown in Tab. 4, we observe a clear advantage of dLMs: at the 1.5B and 4B scales, Efficient-DLM outperforms AR Qwen models of the same sizes by 7.71% and 9.91% on average, respectively. These results also highlight the broader promise of dLMs for other sequence modeling tasks that require bidirectional information.

*Table 4.* Comparing our Efficient-DLM and AR Qwen models on text embedding tasks (Muennighoff et al., 2022).

| Model | Retr. | Ranking | Clust. | Pair Class. | Class. | STS | Avg. |
|---|---|---|---|---|---|---|---|
| Qwen2.5-1.5B | 20.69 | 40.01 | 21.42 | 24.59 | 31.22 | 39.33 | 29.54 |
| Efficient-DLM-1.5B | 18.67 | 43.67 | 23.58 | 56.76 | 31.70 | 49.14 | 37.25 |
| Qwen3-4B | 19.46 | 39.90 | 21.77 | 33.94 | 29.13 | 40.56 | 30.79 |
| Efficient-DLM-4B | 20.17 | 45.05 | 23.91 | 65.59 | 42.22 | 47.27 | 40.70 |

### 5.4. Ablation Study

**Contributions of each component.** We have analyzed the impact of each component of Efficient-DLM in Sec. 2–4. We also summarize their impact in Tab. 5, which performs AR-to-dLM conversion on top of Qwen3-4B. We start from the baseline setting (Dream's bidirectional modeling) and progressively add each component. As observed in Tab. 5, proper attention patterns (with appropriate block-size selection), removing token shift, adding position-dependent token masking, and longer training all contribute to successful AR-to-dLM conversion.

**Benchmark with SOTA dLMs plus Fast-dLLM acceleration.** Existing public dLMs, LLaDA (Nie et al., 2025)

*Table 5.* Ablation study of different AR-to-dLM components by progressively adding each component on the baseline setting.

| Setting | HumanEval | HumanEval+ | MBPP | MBPP+ | GSM8K | Minerva | Avg |
|---|---|---|---|---|---|---|---|
| Bidirectional | 39.02 | 32.32 | 39.60 | 50.00 | 67.40 | 39.17 | 44.59 |
| + Block-wise attention | 53.66 | 50.36 | 55.60 | 69.70 | 78.39 | 46.33 | 59.01 |
| + Remove token shift | 56.10 | 51.22 | 54.60 | 69.84 | 82.87 | 47.02 | 60.27 |
| + Pos-dep masking | 60.37 | 54.27 | 59.00 | 71.43 | 81.12 | 45.92 | 62.02 |
| + Scale to 300B tokens | 60.98 | 56.71 | 60.00 | 70.63 | 86.43 | 49.54 | 64.05 |

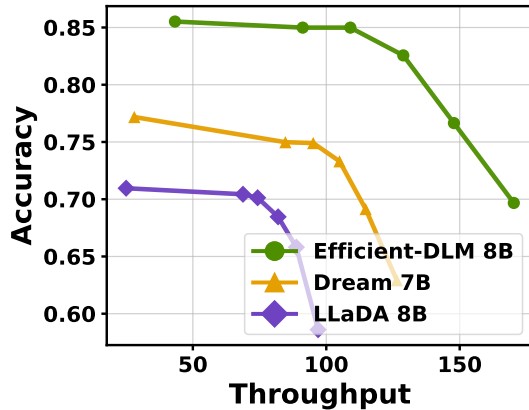

*Figure 10.* Comparing the accuracy-throughput trade-off with Dream/LLaDA plus Fast-dLLM.

and Dream (Ye et al., 2025), both equipped with fully bidirectional attention, can be accelerated by Fast-dLLM (Wu et al., 2025) through partial KV caching and parallel decoding. We benchmark our Efficient-DLM-8B against Dream and LLaDA enhanced with Fast-dLLM's dual cache and parallel decoding, using different confidence thresholds to control the accuracy–throughput trade-off. As shown in Fig. 10, our Efficient-DLM-8B consistently achieves a better accuracy–throughput trade-off than Dream and LLaDA on GSM8K, demonstrating that Efficient-DLM constitutes a stronger dLM family.

## 6. Related Work

**Diffusion language models.** To overcome the token-by-token decoding nature of AR LMs, diffusion LMs, both continuous (Li et al., 2022; Gong et al., 2022; Han et al., 2022) and discrete (Austin et al., 2021; He et al., 2022; Sahoo et al., 2024; Lou et al., 2023; Ou et al., 2024), have been proposed to perform non-AR decoding and thus enable parallel token generation. Among them, masked dLMs (He et al., 2022; Sahoo et al., 2024; Nie et al., 2025; Ye et al., 2025) have been successfully scaled up (e.g., LLaDA (Nie et al., 2025) and Dream (Ye et al., 2025)). Follow-up work has further explored alternative dLM paradigms (Sahoo et al., 2025b;a; Xue et al., 2025), scaled them to larger

generalists (Google DeepMind, 2025) or domain-specific specialists such as coding agents (Song et al., 2025; Gong et al., 2025b; Xie et al., 2025), explored dedicated reinforcement learning schemes (Zhao et al., 2025; Zhu et al., 2025), and extended to more modalities (You et al., 2025).

**Diffusion language model acceleration.** Despite the potential of large dLMs (Nie et al., 2025; Ye et al., 2025), the gap between bidirectional attention and KV caching, along with the one-token-per-step denoising process, limits their achievable speed-up. To address these challenges, dedicated caching strategies for dLMs (Liu et al., 2025; Ma et al., 2025; Wu et al., 2025) have been developed to reuse computations and approximate bidirectional attention. In addition, to realize the potential of parallel token generation, confidence-based sampling (Wu et al., 2025), guidance from AR models (Israel et al., 2025), and adaptive decoding with certainty and positional priors (Wei et al., 2025) have been proposed. Beyond these training-free methods, Block Diffusion (Arriola et al., 2025) combines AR and diffusion by performing block-wise AR and in-block diffusion to support native KV caching and concurrent works (Wu et al., 2025; Cheng et al., 2025; Wang et al., 2025) also convert pretrained AR/dLMs into block-wise dLMs.

## 7. Conclusion

This work systematically explores how to convert pretrained AR models into dLMs that achieve faster generation while retaining strong accuracy. By introducing a continuous pretraining scheme with a block-wise attention pattern, along with a position-dependent token masking strategy that narrows the training–test gap, we provide a principled framework for delivering dLMs with both strong accuracy and speed, resulting in the Efficient-DLM model family. Through comprehensive analyses of attention patterns, training dynamics, and other design choices, our findings offer actionable insights that we hope will guide the community toward building efficient and scalable dLMs. Beyond serving as a practical recipe for AR-to-dLM conversion, our results highlight the broader opportunity to rethink pretraining, masking, and decoding strategies for dLMs in order to realize their promise as alternatives to AR models.

## Impact Statement

This work improves the efficiency of dLMs through AR-to-dLM conversion, reducing inference cost and energy consumption while preserving task performance. This can help make dLMs more accessible and environmentally sustainable. As with other generative modeling advances, increased efficiency may also lower barriers to large-scale deployment, which carries potential risks such as misuse for misinformation or automated content generation. These concerns are not unique to our approach, and we encourage continued development of responsible deployment practices and safeguards alongside progress in model efficiency.

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

*Table 6.* Benchmarking against SOTA AR models and dLMs on 12 tasks spanning coding, math, factual knowledge, and commonsense reasoning. TPF denotes tokens per forward, and TPS refers to throughput measured on an NVIDIA H100 GPU with a batch size of 1. This table is a complement to Tab. 3.

| Type | Model | TPF | TPS (tok/sec) | HumanEval | Coding HumanEval Plus | MBPP | MBPP Plus | GSM8K | Math Minerva Math | Factual MMLU | ARC-E | Commonsense Reasoning ARC-C | Hellaswag | PIQA | Winogrande | Avg |
|---|---|---|---|---|---|---|---|---|---|---|---|---|---|---|---|---|
| AR | Llama-3.2-1B | 1.00 | 143.91 | 17.68 | 14.63 | 26.60 | 38.89 | 5.69 | 4.28 | 30.98 | 65.28 | 36.35 | 63.76 | 74.43 | 63.30 | 36.82 |
| | Qwen2.5-0.5B | 1.00 | 99.93 | 27.44 | 25.61 | 29.60 | 44.97 | 37.45 | 14.48 | 47.65 | 64.77 | 31.83 | 52.25 | 70.02 | 57.70 | 41.98 |
| | Qwen2.5-1.5B | 1.00 | 73.03 | 35.98 | 29.88 | 43.60 | 59.23 | 64.97 | 28.98 | 60.96 | 75.17 | 45.05 | 67.90 | 76.12 | 65.75 | 54.47 |
| dLM (Ours) | Efficient-DLM-1.5B | 2.33 | 158.89 | 41.46 | 35.37 | 38.80 | 53.70 | 57.92 | 26.64 | 57.63 | 75.00 | 45.31 | 59.43 | 71.44 | 61.72 | 52.04 |
| | | 2.69 | 184.48 | 41.46 | 35.98 | 37.60 | 52.12 | 58.07 | 25.52 | 57.63 | 75.00 | 45.31 | 59.43 | 71.44 | 61.72 | 51.77 |
| AR | Qwen3-1.7B | 1.00 | 71.59 | 48.17 | 41.46 | 55.80 | 71.43 | 73.54 | 34.76 | 62.53 | 73.32 | 44.62 | 66.43 | 75.63 | 64.96 | 59.39 |
| | Qwen3-4B | 1.00 | 47.13 | 57.32 | 50.61 | 66.80 | 80.69 | 85.44 | 47.10 | 73.19 | 79.12 | 51.62 | 73.66 | 78.07 | 72.06 | 67.97 |
| | Qwen3-8B | 1.00 | 42.51 | 64.63 | 56.71 | 69.40 | 83.07 | 86.73 | 52.94 | 76.93 | 81.90 | 53.16 | 78.59 | 79.22 | 75.69 | 71.58 |
| dLM | LLaDA-8B | 1.00 | 25.04 | 32.32 | 27.44 | 40.80 | 51.85 | 70.96 | 27.30 | 65.86 | 73.78 | 49.15 | 71.05 | 73.88 | 74.66 | 54.92 |
| | Dream-7B | 1.00 | 28.11 | 54.88 | 49.39 | 56.80 | 74.60 | 77.18 | 39.60 | 67.00 | 82.20 | 59.13 | 73.73 | 75.52 | 73.56 | 65.30 |
| dLM (Ours) | Efficient-DLM-4B | 2.52 | 119.33 | 61.59 | 56.71 | 57.60 | 69.58 | 87.57 | 49.56 | 71.80 | 81.52 | 55.80 | 69.02 | 75.46 | 72.53 | 67.39 |
| | | 3.01 | 130.24 | 60.98 | 56.10 | 57.20 | 69.58 | 87.19 | 49.02 | 71.80 | 81.52 | 55.80 | 69.02 | 75.46 | 72.53 | 67.18 |
| dLM (Ours) | Efficient-DLM-8B | 1.00 | 39.99 | 68.29 | 63.41 | 61.00 | 76.72 | 88.32 | 50.12 | 77.22 | 84.89 | 61.86 | 72.53 | 77.53 | 77.58 | 71.62 |
| | | 2.74 | 109.78 | 64.02 | 59.15 | 63.20 | 76.19 | 86.73 | 50.30 | 77.22 | 84.89 | 61.86 | 72.53 | 77.53 | 77.58 | 70.93 |
| | | 3.27 | 130.71 | 63.41 | 58.54 | 62.20 | 75.66 | 86.35 | 50.06 | 77.22 | 84.89 | 61.86 | 72.53 | 77.53 | 77.58 | 70.65 |

# A. Detailed Experimental Settings

**Evaluation settings.** For all evaluations of our Efficient-DLM in Sec. 5.1, we follow the best practice from Sec. 2.3 and use evaluation block sizes of 16 for Efficient-DLM-1.5B and 32 for Efficient-DLM-4B/8B, respectively. We use lm-evaluation-harness (Gao et al., 2024) to evaluate AR baselines (Qwen3 (Yang et al., 2025), Qwen2.5 (Team, 2024), Llama-3.2 (Grattafiori et al., 2024)); for dLMs (LLaDA (Nie et al., 2025) and Dream (Ye et al., 2025)), we adopt their official evaluation code. We benchmark 12 tasks covering math (GSM8K, Minerva Math), coding (HumanEval, HumanEval Plus, MBPP, MBPP Plus), factual knowledge (MMLU), and commonsense reasoning (ARC-C, ARC-E, Hellaswag, PIQA, Winogrande). Following Dream (Ye et al., 2025), we use 8-shot, 4-shot, 0-shot, 0-shot, 3-shot, and 3-shot settings for GSM8K, Minerva Math, HumanEval, HumanEval Plus, MBPP, and MBPP Plus, respectively. The maximum number of generated tokens is set to 512 for all tasks, except GSM8K, which uses 256 as in (Ye et al., 2025).

**Parallel decoding settings.** Following (Wu et al., 2025), we set a confidence threshold and decode all tokens that exceed the threshold at each denoising step to enable parallel decoding. Tokens per forward (TPF) and generation throughput (tok/sec), reported in Tab. 3, are averaged over all six generation tasks.

**Text embedding evaluation settings.** Following the ablation setting of LLM2Vec (BehnamGhader et al., 2024), we evaluate the models on six categories of tasks from MTEB (Muennighoff et al., 2022), including retrieval (SciFact, ArguAna, NF-Corpus), reranking (StackOverflowDupQuestions, SciDocsRR), clustering (BiorxivClusteringS2S, MedrxivClusteringS2S, TwentyNewsgroupsClustering), pair classification (SprintDuplicateQuestions), classification (Banking77Classification, EmotionClassification, MassiveIntentClassification), and semantic textual similarity (STS17, SICK-R, STSBenchmark). In total, the evaluation covers 15 datasets.

To obtain sequence embeddings, we apply mean pooling over the last layer's hidden states across all tokens. For Qwen models, we use a causal attention mask, as switching to a bidirectional mask consistently degraded performance. For our Efficient-DLM models, we instead employ a bidirectional attention mask. All experiments are conducted in a zero-shot setting, directly using the pretrained weights of Qwen and Efficient-DLM without any fine-tuning.

# B. Per-task Accuracy Achieved by Efficient-DLM and Baselines

We provide the per-task accuracy achieved by Efficient-DLM and SOTA AR/dLMs in Tab. 6 as a complement to Tab. 3 of our main paper.

# C. More Benchmarks with SOTA AR LMs and dLMs

**The accuracy–throughput trade-off with larger inference batch sizes.** As a complement to Sec. 5.2, we further visualize the trade-off between accuracy and throughput under different batch sizes for multiple models on the GSM8K dataset. As shown in Fig. 11, we observe that (1) our Efficient-DLM-8B consistently improves the accuracy–efficiency trade-off compared to both AR models (Qwen3-1.7B–8B) and dLMs, up to a batch size of 16; and (2) the efficiency benefits of dLMs over AR models are more pronounced at small batch sizes, which correspond to more memory-bounded scenarios, and these benefits begin to diminish at larger batch sizes, e.g., Efficient-DLM-8B falls behind Qwen3-1.7B in throughput at a batch size of 32.

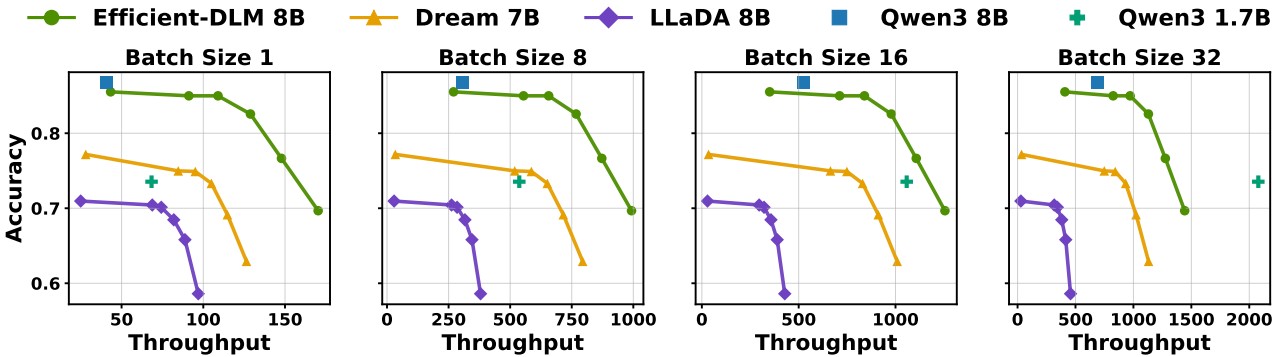

*Figure 11.* Visualizing the accuracy-throughput trade-off under different batch sizes on the GSM8K dataset.

This set of experiments highlights the current limitations of dLMs in large-batch serving scenarios. Potential workarounds include adaptive block sizes, improved parallel sampling schemes, and combining dLMs with linear attention to enhance large-batch efficiency, which we leave for future work.

## D. The Impact of Initial LR

When initializing from pretrained AR models, the learning rate for continuous pretraining is a key hyperparameter, as it controls the speed of weight changes that affect both the preservation of the pretrained models' abilities and the adaptation to dLMs' new attention patterns. We perform an ablation study on Qwen3-4B trained for 25B tokens with different initial learning rates using a cosine learning rate schedule.

**Observations and analysis.** As shown in Tab. 7, we find that there exists a sweet-spot learning rate setting, e.g., 1e-5 in our case, that balances both aspects mentioned above. Intuitively, overly large learning rates cause greater weight drifts and degrade the pretrained models' original abilities, while overly small learning rates cannot effectively adapt to the new attention pattern. We also note that for any design factors in continuously training a pretrained model into dLMs, these two aspects should be carefully balanced to achieve decent final accuracy. Based on this set of experiments, we adopt 1e-5 as the default initial learning rate throughout the main manuscript.

*Table 7.* Results of continuous pretraining with different initial learning rates on Qwen3-4B for 25B tokens.

| Init LR | HumanEval | HumanEval Plus | MBPP | MBPP Plus | GSM8K | Minerva Math | Avg |
|---|---|---|---|---|---|---|---|
| 1.00E-04 | 49.39 | 43.90 | 44.20 | 56.08 | 72.56 | 39.54 | 50.95 |
| 3.00E-05 | 54.27 | 49.39 | 52.40 | 67.46 | 77.48 | 38.56 | 56.59 |
| 1.00E-05 | 57.93 | 51.22 | 54.40 | 71.96 | 81.73 | 46.54 | 60.63 |
| 3.00E-06 | 56.10 | 50.61 | 54.60 | 67.99 | 83.93 | 47.44 | 60.11 |
| 1.00E-06 | 45.73 | 42.68 | 47.20 | 66.67 | 81.12 | 43.94 | 54.56 |

## E. Loss Distributions across Tokens

To study the difference in loss distributions across token positions between AR models and dLMs, we visualize the training loss defined in Eq. 1 for diffusion Qwen2.5-1.5B trained with a block size of 16, alongside the AR Qwen2.5-1.5B trained with an AR loss.

As shown in Fig. 12, which shows the average loss of the first 256 tokens in training sequences, we observe that (1) in AR models, the initial tokens incur higher loss due to the lack of context, while the loss of later tokens becomes more uniform; and (2) in dLMs, the loss follows a periodic pattern aligned with block boundaries, where later tokens within each block show higher loss due to limited clean context, consistent with Fig. 6 (c). In addition, similar to AR models, the initial tokens of the entire sequence also experience higher loss from insufficient context.

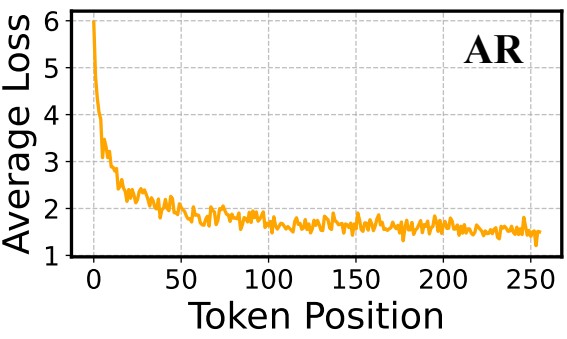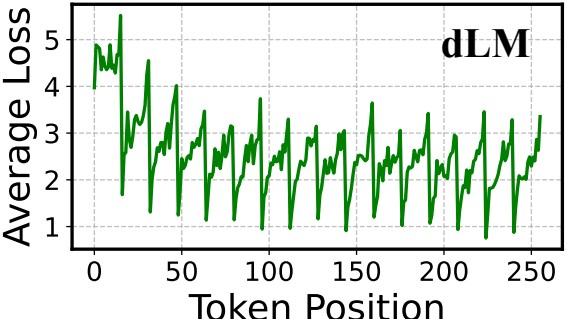

*Figure 12.* Visualizing the loss distributions over token positions of AR models and dLMs.

## F. AR-to-dLM Conversion via Parameter-Efficient Tuning

Motivated by the relatively small weight changes observed in Sec. 2.2, we investigate whether parameter-efficient tuning can effectively convert pretrained AR models into dLMs. To this end, we apply Low-Rank Adaptation (LoRA) (Hu et al., 2022) to all linear layers in attention/FFN modules, combined with the best attention pattern identified in Sec. 2.2, i.e., block-wise attention conditioning on clean context without token shift. All other parameters are frozen, except for the embedding layer, normalization operators, and the final model head, which we find must remain trainable for effective adaptation.

**Observations and analysis.** We extend Tab. 1 into Tab. 8 by including LoRA tuning results with two different ranks in Rows (h) and (i). We observe that LoRA tuning achieves reasonably good performance for AR-to-dLM conversion. Specifically, LoRA with rank 64 (Row i) surpasses the full-model training results of fully bidirectional attention and block-wise attention without clean context, while remaining 7.63% behind the full-model training results of the best scheme, i.e., block-wise attention with clean context. These results indicate that (1) with proper training schemes, even parameter-efficient tuning can yield competitive dLMs, and (2) full-model training remains necessary to obtain strong dLMs.

*Table 8.* Comparison of different dLM training schemes on Qwen2.5-1.5B. This table extends Tab. 1, with Rows (h) and (i) added to present the LoRA tuning results.

| Row ID | Attn Pattern | Clean Context | Token Shift | KV Cache | LoRA Rank | Human -Eval | Human -Eval Plus | MBPP | MBPP Plus | GSM8K | Minerva Math | Avg |
|---|---|---|---|---|---|---|---|---|---|---|---|---|
| a | AR | - | ✔ | ✔ | - | 36.59 | 29.88 | 43.6 | 59.52 | 54.74 | 26.40 | 41.79 |
| b | Bidirectional | - | ✔ | ✘ | - | 15.85 | 12.20 | 16.2 | 24.34 | 28.96 | 11.08 | 18.10 |
| c | Bidirectional | - | ✘ | ✘ | - | 19.51 | 15.24 | 17.2 | 24.34 | 28.20 | 11.22 | 19.29 |
| d | Block-wise | ✘ | ✔ | ✔ | - | 31.10 | 25.61 | 23.6 | 36.77 | 38.44 | 13.88 | 28.23 |
| e | Block-wise (2×) | ✘ | ✔ | ✔ | - | 26.22 | 22.56 | 26.0 | 42.33 | 36.69 | 12.56 | 27.73 |
| f | Block-wise | ✔ | ✔ | ✔ | - | 38.41 | 33.54 | 33.0 | 48.68 | 51.48 | 21.04 | 37.69 |
| g | Block-wise | ✔ | ✘ | ✔ | - | 39.02 | 34.76 | 34.0 | 48.15 | 52.99 | 21.56 | 38.41 |
| h | Block-wise (LoRA) | ✔ | ✘ | ✔ | 16 | 30.49 | 25.61 | 20.60 | 30.95 | 43.82 | 16.08 | 27.93 |
| i | Block-wise (LoRA) | ✔ | ✘ | ✔ | 64 | 28.66 | 25.61 | 24.40 | 40.21 | 48.14 | 17.64 | 30.78 |

