# OpenReview forum: "Efficient-DLM: From Autoregressive to Diffusion Language Models, and Beyond in Speed"
_ICML.cc/2026/Conference — ICML 2026 regular_

### Official Review · Reviewer_rAix · 2026-02-26

**Soundness:** 3
**Presentation:** 3
**Significance:** 2
**Originality:** 3
**Overall Recommendation:** 4
**Confidence:** 4

**Summary:**

The main contribution of the paper is that it analyzes the generation tendency and finds that diffusion language models tend to generate from left to right, with left tokens having higher confidence.
Given this prior, the author proposes tuning the noise pattern during training, making the tokens on the left less likely to be masked while those on the right more likely to be masked.
This mask strategy helps bridge the train-test gap and improve task performance.

**Compliance With Llm Reviewing Policy:**

Affirmed.

**Final Justification:**

Concern addressed. Score raised from 3->4.

**Key Questions For Authors:**

1. What's the max generation length when evaluating the AR baselines? In my experience, the AR model requires a longer generation length than the dLMs trained from the AR model, because the training max-length setting is different during instruction training. For example, the 4-shot minerva math performance in Qwen2.5-1.5B AR baseline is lower than the 0-shot AR performance in [1], and the 8-shot GSM8K performance in Qwen3-8B AR baseline is lower than the 4-shot AR performance in [2]. The gaps in the evaluation between dLMs studies are quite confusing.

[1] Wu, Chengyue, et al. "Fast-dllm v2: Efficient block-diffusion llm." arXiv preprint arXiv:2509.26328 (2025).
[2] Cheng, Shuang, et al. "Sdar: A synergistic diffusion-autoregression paradigm for scalable sequence generation." arXiv preprint arXiv:2510.06303 (2025).

**Limitations:**

yes

**Strengths And Weaknesses:**

1. Soundness: The submission is technically sound. The from-left-to-right generation tendency is reasonable and is also found in some recent studies.
2. Presentation: The submission is clearly written and well structured. The highlighted takeaways help read the paper.
3. Significance: The submission is studying an important issue in diffusion language models, which is the masking strategy. However, one important mask strategy: complementary masking[1] is not evaluated in this paper. What's worse, the from-left-to-right masking strategy will becomes from-right-to-left masking strategy in the complementary masking settings, which means that the methods in the current paper is not orthogonal to [1].
4. Originality: Although I agree that the generation tendency is novel, other points in the paper such as "Block-wise is better when transforming from AR to dLMs", "Block size matters", "Block size can be different during training and inference" are already mentioned in other studies[1,2,3], should not take up too much space in the paper.

[1] Wu, Chengyue, et al. "Fast-dllm v2: Efficient block-diffusion llm." arXiv preprint arXiv:2509.26328 (2025).
[2] Cheng, Shuang, et al. "Sdar: A synergistic diffusion-autoregression paradigm for scalable sequence generation." arXiv preprint arXiv:2510.06303 (2025).
[3] Song, Yuxuan, et al. "Seed diffusion: A large-scale diffusion language model with high-speed inference." arXiv preprint arXiv:2508.02193 (2025).

---

> ### Author Rebuttal · Authors · 2026-03-31
>
> Thank you for recognizing the significance of our target problem and the soundness of our techniques, as well as for your constructive comments! We have answered all your questions and concerns as follows:
>
> **Q1. Clarifications on the training/evaluation settings**
>
> We would like to clarify that all experiments in our work are base models trained with continuous pretraining, rather than instruction tuning. In contrast, the reported results in Fast-dLLM v2 / SDAR that you mentioned are all based on instruction-tuned models, so the absolute values across benchmarks are not directly comparable.
>
> For AR evaluation, we use the standard lm-eval-harness to evaluate official Qwen base models on HF, with few-shot settings following Dream/LLaDA for fair comparison with dLMs. We adopt the default generation length in their HF configuration, i.e., 2048. The results of the base models generally match those reported in the Qwen3 Technical Report.
>
> We believe continuous pretraining provides a cleaner setting to study AR-to-DLM conversion, as it is less affected by the choice of chat templates and highly task-specific data in SFT datasets. We hope this clarification addresses your concerns regarding the training and evaluation settings.
>
> ---
>
> **Q2. Experiments on complementary masking**
>
> Thank you for the good question! Our key finding is that complementary masking is less effective in terms of scaling (i.e., the accuracy evolution with training time) under the continuous pretraining setting (as clarified in Q1), where more tokens are available; therefore, we did not use it.
>
> To support this, we performed a study on Qwen2.5 1.5B and Qwen3 4B for AR-to-DLM conversion, each under three settings: (1) 12.5B-token training with *uniform masking*, (2) 25B-token training with *uniform masking*, and (3) 12.5B-token training with *complementary masking*, where the training cost is comparable to 25B-token training with uniform masking since each sequence needs to be trained twice. Note that we did not apply position-dependent masking in any setting here for a fair comparison.
>
> As shown in the table below, we observe that:
>
> (1) Under the same number of unique tokens (12.5B), complementary masking improves the average accuracy by 0.26%–0.86% on both models compared to uniform masking;
>
> (2) However, under the same training time, simply training on more unique tokens (25B) with uniform masking leads to notably better average accuracy (>3%) compared to complementary masking.
>
> As such, under the setting of multi-epoch training on limited data (common in SFT settings), complementary masking might be beneficial; however, when data is richer, we find that simply training on more tokens with uniform masking leads to better accuracy under the same training budget.
>
>
> |**Model**|**Training Setting**|**Training Time**|**HumanEval**|**HumanEval+**|**MBPP**|**MBPP+**|**GSM8K**|**Minerva Math**|**Avg**|
> |:---:|:---:|:---:|:---:|:---:|:---:|:---:|:---:|:---:|:---:|
> ||12.5B + uniform masking|1x|30.49|25.00|29.60|47.88|53.07|21.04|34.51|
> |Qwen2.5 1.5B|12.5B + complementary masking|2x|32.32|27.44|30.40|48.68|51.86|21.52|35.37|
> ||25B + uniform masking|2x|39.02|34.76|34.00|48.15|52.99|21.56|**38.41**|
> ||12.5B + uniform masking|1x|48.78|43.29|48.00|66.40|82.49|48.44|56.23|
> |Qwen3 4B|12.5B + complementary masking|2x|45.73|42.68|52.00|68.78|81.96|47.76|56.49|
> ||25B + uniform masking|2x|56.10|51.22|54.60|69.84|82.87|47.02|**60.28**|
>
>
> ---
>
> **Q3. Originality of our method**
>
> We agree that block-wise attention has been discussed in the works you mentioned (concurrent with our first version manuscript) since the first Block Diffusion work (M. Arriola, ICLR’25). Our major contribution here is to dive deeper into each component, i.e., block-wise attention, clean-context conditioning, and token shift, to study their individual impacts to the AR-to-DLM conversion process and determine the best training recipe. This study also provides the key insight that the key to effective AR-to-dLM conversion is preserving the pretrained AR distribution.
>
> In addition to attention patterns, we also study a missing aspect in prior works, i.e., the training objectives (which tokens to mask during training). We identify the training–test gap in token masking schemes and propose the concept of position-dependent token masking to alleviate it, as well as to inspire future dLM masking strategies;
>
> Furthermore, we study the scaling of dLMs with longer training and find that it results in improved likelihood estimation and better decoding parallelism. We also validate dLMs’ promise for one-for-all flexibility and their advantages in producing stronger text embeddings, which make dLMs promising alternatives to AR in scenarios where these capabilities are important.
>
> We hope the delivered insights and empirical results can inspire future AR-to-dLM training techniques and help guide decisions on when to use dLMs.

---

> > ### Author Rebuttal · Reviewer_rAix · 2026-04-03
> >
> > Raise from 3->4.

---

### Official Review · Reviewer_DRVg · 2026-03-06

**Soundness:** 4
**Presentation:** 4
**Significance:** 3
**Originality:** 3
**Overall Recommendation:** 5
**Confidence:** 4

**Summary:**

The paper proposes a method to efficiently convert pre-trained AR language models into Diffusion Language Models. To achieve this, the authors identify two major limitations in existing AR-to-dLM conversion schemes. First, they show that fully bidirectional attention causes significant weight drift from the pretrained AR initialization, and propose instead a block-wise attention pattern conditioned on clean context (no token shift) which better preserves the pretrained AR weight distribution and enables native KV caching. Second, to address the distribution shift between uniform masking during training and the left-to-right generation tendency observed during confidence-based parallel decoding at inference, they propose a novel position-dependent token masking strategy that assigns higher masking probabilities to later tokens in a block during training. The resulting Efficient-DLMs achieve state-of-the-art accuracy-throughput trade-offs compared to prior dLMs and strong AR baselines in small batch settings, and also demonstrates strong zero-shot performance on text embedding tasks.

**Compliance With Llm Reviewing Policy:**

Affirmed.

**Final Justification:**

The authors solved all my concerns.

**Key Questions For Authors:**

1. In Appendix C (Figure 11), the throughput advantage of Efficient-DLM over AR models drops significantly at a batch size of 32. Could you elaborate on the specific hardware/computational bottlenecks (e.g., memory bandwidth vs. compute) causing this? Are there specific architectural changes or systems-level optimizations that could improve large-batch efficiency for dLMs?
2. To fully isolate the architectural benefits of the dLM from the effect of encountering 300B-500B additional high-quality tokens, how does Efficient-DLM compare to its base AR model continuously pretrained on the exact same token mixture with a standard causal objective?
3. In Section 2.2, removing the "token shift" objective improves performance. Given that the base AR model was trained entirely with token shift, doesn't removing it create an initial objective misalignment? Why does the model adapt to predicting the masked token directly without causing larger initial weight drifts?
4. The optimal half-life ratio for the position-dependent masking was found to be 0.1. How sensitive is the model to the choice of this parameter? Does the optimal lambda change depending on the model scale or the chosen training block size?

**Limitations:**

yes

**Strengths And Weaknesses:**

Strengths:
1. The paper provides a highly systematic investigation into AR-to-dLM conversion. The ablation studies isolating the effects of attention patterns, clean context, and token shift on weight drift provide solid scientific justification for their design choices. The takeaways throughout the paper offer actionable insights for the community.
2. The identification of the training-inference gap in token masking and the proposed position-dependent masking strategy is an intuitive and effective solution that aligns the training objective with the autoregressive nature of language observed during parallel decoding.
3. The proposed Efficient-DLM models demonstrate impressive performance, pushing the accuracy-throughput Pareto frontier beyond both strong AR baselines and recent dLMs at a batch size of 1. The strong performance on bidirectional text embedding tasks is also a nice addition that highlights the versatility of the converted models.

Weaknesses:
1. A significant practical limitation of the proposed method (and dLMs in general) is that the throughput advantages over AR models heavily diminish or reverse at larger batch sizes (e.g., BS=32), as shown in Figure 11. Since high-throughput production serving relies heavily on large batching, this restricts the immediate practical impact of the "Beyond in Speed" claim primarily to latency-sensitive, small batch scenarios.
2. The "continuous pretraining" phase still requires a massive computational budget (300B to 500B tokens, Table 5). While this is more efficient than training a dLM from scratch, it represents a very high computational barrier that blurs the line between rapid conversion and full pre-training.
3. Given the massive 300B-500B token budget, the paper does not compare the final generative performance against an AR model that is simply continuously pre-trained on the exact same token mixture using a standard causal objective. This baseline is needed to isolate the architectural benefits of the dLM from the gains of seeing additional high-quality tokens.

---

> ### Author Rebuttal · Authors · 2026-03-31
>
> Thank you for recognizing the systematic nature of our study and the impressiveness of our results, as well as for your constructive comments! We have answered all your questions as follows:
>
> **Q1. The large-batch efficiency of dLMs**
>
> Please refer to our response to Q4 of Reviewer tuek.
>
> ---
>
> **Q2. Computation cost of AR-to-Diffusion conversion**
>
> Please refer to our response to Q3 of Reviewer tuek.
>
> ---
>
> **Q3. Efficient-DLM vs. AR counterparts trained on the same training tokens**
>
> Thank you for the good question! We also believe this experiment is important to include. Following your suggestion, we continuously pretrain the official Qwen3 4B using a pure AR objective on the same data mixture and the same number of tokens (300B), and compare it with the starting-point AR model.
>
> As shown in the table below, we observe that since the official Qwen3 4B has already been pretrained on 36T high-quality tokens, further continuous pretraining on our data mixture improves task accuracy only to a limited extent (+0.78% accuracy on average).
>
> This indicates that the performance gains of our Efficient-DLM family mainly come from algorithmic improvements that recover AR accuracy, rather than being over-boosted by high-quality datasets. We will add more analysis in the final version.
>
> |**Setting**|**HumanEval**|**HumanEval+**|**MBPP**|**MBPP+**|**GSM8K**|**Minerva Math**|**MMLU**|**ARC-E**|**ARC-C**|**Hellaswag**|**PIQA**|**Winogrande**|**Avg**|
> |:---:|:---:|:---:|:---:|:---:|:---:|:---:|:---:|:---:|:---:|:---:|:---:|:---:|:---:|
> |Official Qwen3 4B|57.32|50.61|66.80|80.69|85.44|47.10|73.19|79.12|51.62|73.66|78.07|72.06|67.97|
> |Qwen3 4B + Continuous Pretraining|59.76|54.88|63.80|77.87|82.56|48.14|74.01|83.04|57.25|72.29|78.84|72.61|68.75|
> |Efficient-DLM-4B|60.98|56.71|60.00|70.63|86.43|49.54|71.80|81.52|55.80|69.02|75.46|72.53|67.54|
>
>
> ---
>
> **Q4. More analysis on token shift**
>
> This is a good question. We consistently find that token shift is unnecessary across settings, and we provide our hypothesis in Section 2.2 of our manuscript.
>
> Our assumption is that directly inheriting token shift does not mean inheriting the same training objective as AR, since AR objectives are defined over fully clean tokens, while the dLM objective is defined over masked tokens. More specifically, with token shift, predicting the next token of a masked token involves two steps: inferring the masked token itself and predicting the next token. This could be harder than simply inferring the masked token when no token shift is applied.
>
> As such, our view is that the inherited AR abilities mainly affect how the clean context is encoded, while dLMs learn to decode masked tokens based on the context. Empirically, not applying token shift to masked token outputs works better, as it is potentially a simpler task to learn.
>
> ---
>
> **Q5. Ablation study on the optimal lambda in position-dependent masking**
>
> We have provided an ablation study on lambda in Table 2 of our submitted manuscript, where we observe that introducing lambda within a reasonable range generally boosts generation accuracy. Following your suggestion, in addition to Table 2 in our manuscript, we further apply position-dependent masking in two additional settings: (1) AR-to-diffusion conversion on Ministral3 8B for 25B tokens, which has a different architecture, scale, and pretraining recipe from Qwen3 4B, and (2) a 4B model (with the same architecture as Qwen3 4B) trained from scratch.
>
> As shown in the table below, we observe that the three lambda choices we experimented with consistently bring improvements over the baseline across both AR-to-DLM conversion on Ministral3 8B and the 4B training-from-scratch setting. For example, under the training-from-scratch setting, position-dependent masking can bring up to a 20% average accuracy improvement. We assume the large improvement is because training on all permutations equally from scratch (without any prior from AR initialization) is difficult, and introducing left-to-right linguistic priors is particularly helpful in this case.
>
> This set of experiments indicates that the same set of lambda choices is generally suitable across diverse settings and does not require extensive tuning.
>
>
> |Model Setting|Lambda Setting|HumanEval|HumanEval+|MBPP|MBPP+|GSM8K|Minerva Math|Avg|
> |:---:|:---:|:---:|:---:|:---:|:---:|:---:|:---:|:---:|
> ||Baseline (no pos-dep masking)|58.54|52.44|53.00|73.81|83.17|55.84|62.80|
> |Setting 1: AR-to-DLM|Lambda=0.05|64.63|54.88|57.00|73.55|78.77|55.62|64.07|
> |@Ministral3 8B|Lambda=0.1|60.98|53.05|60.80|74.87|83.17|56.14|64.83|
> ||Lambda=0.25|63.41|57.93|57.80|70.90|81.12|60.54|65.28|
> ||Baseline (no pos-dep masking)|14.02|12.80|14.40|23.02|19.71|11.21|15.86|
> |Setting 2: 4B dLM|Lambda=0.05|32.93|30.49|26.60|44.44|46.32|21.06|33.64|
> |Trained from Scratch|Lambda=0.1|34.76|31.71|28.80|44.18|52.62|23.53|35.93|
> ||Lambda=0.25|33.54|30.49|31.20|47.62|53.07|24.72|36.77|

---

> > ### Author Rebuttal · Reviewer_DRVg · 2026-04-02
> >
> > The authors have addressed all of my concerns, so I will maintain my current score.

---

### Official Review · Reviewer_mprB · 2026-03-12

**Soundness:** 3
**Presentation:** 3
**Significance:** 3
**Originality:** 3
**Overall Recommendation:** 4
**Confidence:** 3

**Summary:**

The core contribution of this paper is the proposal of a continuous pre-training framework for efficiently converting pre-trained autoregressive models into diffusion language models. Methodologically, it employs block-wise attention, clean context, token shift removal, and position-dependent masking, aiming to improve parallel generation throughput while preserving the original AR capabilities as much as possible. Experimentally, the authors not only performed complete ablation but also presented the final Efficient-DLM family; the 8B model achieves a 5.35% improvement in average accuracy and a 4.50× improvement in throughput compared to Dream 7B, and also shows a 2.68% improvement in accuracy and a 2.77× improvement in throughput compared to Qwen3 4B.

**Compliance With Llm Reviewing Policy:**

Affirmed.

**Final Justification:**

The authors have addressed all of my concerns, so I will maintain my positive score.

**Key Questions For Authors:**

1. Is this method robust to "non-Quadrical initialization"? Block-wise causality is closer to AR and can reduce weight drift, which is the core argument of the paper; however, different AR models may have different pre-training distributions, position encodings, tokenizers, and data formulations, and whether they will all achieve the same transformation benefit has not yet been systematically verified.

2. Will the block-wise structure create a new upper limit? The authors proved that it is better suited to starting from AR initialization than fully bidirectional, but this does not automatically mean that it is the final optimal structure for dLM. Is block-wise a local optimum that is "better suited for transformation," or does it still outperform stronger bidirectional structures under sufficiently large-scale training? This question has not yet been fully answered.

**Limitations:**

yes

**Strengths And Weaknesses:**

**Advantages**

1. The motivation behind the method is very clear, and the modifications are targeted and effective.
The authors first point out two key problems with existing AR-to-dLM conversions: fully bidirectional attention leads to greater weight drift, excessive context contamination, and difficulties in KV caching; uniform masking does not match the left-to-right generation trend based on confidence during testing. The proposed block-wise attention with clean context and position-dependent masking address these two problems respectively, forming a complete methodological chain.

2. The experimental coverage is comprehensive, examining both accuracy and parallel generation capabilities.
The authors not only evaluated 12 generation/inference tasks but also specifically analyzed factors such as training block size, evaluation block size, TPF/NFE, and training dynamics; in particular, Section 4 emphasizes that dLM evaluation should not only consider the accuracy of "one token per step" but also the parallel generation capability under low NFE, a valuable perspective.

3. Position-dependent masking is both intuitive and empirically supported.
The authors observed a clear left-to-right stabilization trend in tokens during testing, thus allowing later tokens to have a higher mask probability in the later stages of denoising during training. In the experiments, λ=0.1 improved performance by an average of 2.61% compared to uniform masking, and the benefit was even greater under more aggressive parallel decoding, indicating that this is not only effective at a single setting.

**Disadvantages**

1. The method is still relatively "engineering-driven," with many hyperparameters. Block size, whether the context is clean, whether shifting is used, the masking λ, and the confidence threshold during inference all significantly affect the results. Although the paper analyzes these factors, it currently resembles an empirically effective recipe rather than a more unified and predictable theoretical framework.

2. The diversity of experimental backbones is insufficient. The core experiments and the final model are almost entirely based on Qwen2.5/Qwen3 initialization, thus the paper strongly proves that "this conversion scheme is effective under Qwen-based initialization," but the evidence for its applicability to other AR families is insufficient.

---

> ### Author Rebuttal · Authors · 2026-03-31
>
> Thank you for recognizing the valuableness of our findings, as well as for your constructive comments! We have answered all your questions as follows:
>
> **Q1. Novelty & “engineering-driven” with hyperparameters**
>
> We agree that the AR-to-DLM conversion process involves many hyperparameters, but it is precisely our goal to understand the dynamics behind them and provide a practical guideline with actionable insights to simplify their selection, which is an important step in evaluating dLMs as alternatives to AR LMs. Concretely, we make three key contributions to the community:
>
> (1) *Attention design:* We systematically compare attention patterns and show that block-wise attention best balances diffusion flexibility with block-level causality, which preserves the pretrained AR distribution.
>
> (2) *Training objective:* We identify a training–test mismatch in token masking and propose position-dependent masking to mitigate this gap and guide future designs.
>
> (3) *Scaling insights:* We show that longer training improves likelihood and decoding parallelism, and we validate dLMs’ one-for-all flexibility and stronger embedding quality in relevant scenarios.
>
> Overall, we hope these insights and empirical results can guide when and how to effectively adopt dLMs.
>
> ---
>
> **Q2. Experiments on more model backbones**
>
> Thank you for the helpful suggestion! We extend our framework to another AR model family, Ministral3 8B, to validate its effectiveness across different architectures and pretraining setups, and we will further expand to more models in future work.
>
> Concretely, we progressively apply our techniques during AR-to-DLM conversion: starting from bidirectional attention, then introducing block-wise attention, and finally adding position-dependent masking (all 25B tokens). As shown in the table below, we observe consistent trends with our main results:
>
> (1) Block-wise attention provides clear gains over bidirectional attention;
>
> (2) Position-dependent masking further improves accuracy by introducing left-to-right linguistic priors.
>
> These results support the generality of our insights. We will include this discussion in the final version.
>
> |Setting|HumanEval|HumanEval+|MBPP|MBPP+|GSM8K|Minerva Math|Avg|
> |:---:|:---:|:---:|:---:|:---:|:---:|:---:|:---:|
> |bidirectional|50.61|46.95|52.20|70.63|68.46|43.72|55.43|
> |\+block-wise attention|58.54|52.44|53.00|73.81|83.17|55.84|62.80|
> |\+pos-dependent masking|60.98|53.05|60.80|74.87|83.17|56.14|64.83|
>
> ---
>
> **Q3. Whether block-wise attention is a local optimum in larger-scale training**
>
> This is an insightful question. Our view is that although block-wise attention may not necessarily be the single optimal structure for dLMs, it can outperform bidirectional attention across settings. This is because, in addition to being closer to AR initialization and thus better inheriting pretrained abilities, block-wise causality also better aligns with the left-to-right linguistic priors of language and is more suitable for language modeling tasks. (Additionally, block-wise attention has the benefit of KV caching.)
>
> To verify this, we further compare block-wise and bidirectional attention under three settings: (1) training a 4B dLM (with the same architecture as Qwen3 4B) from scratch with 600B tokens, (2) AR-to-DLM conversion with 25B tokens on Qwen3 4B, and (3) AR-to-DLM conversion with 25B tokens on Ministral3 8B (different scale, architecture, and pretraining recipes from Qwen3 4B).
>
> From the table, we observe that:
>
> (1) When trained from scratch, block-wise attention also achieves better accuracy than bidirectional attention. This indicates that even without pretrained AR initialization, block-wise attention is natively a better attention pattern than bidirectional ones, potentially because it aligns with left-to-right linguistic priors through block-wise causality.
>
> (2) For AR-to-DLM conversion across model archs / scales, the benefit of block-wise attention over the bidirectional one is consistent.
>
> These experiments indicate that both benefits of block-wise attention, preserving AR initialization and aligning with linguistic priors, lead to better results than bidirectional attention. As such, we view block-wise attention as a better local optimum than bidirectional attention in the development of dLMs, and it could inspire more advanced attention patterns.
>
> |**Setting**|**Attn Pattern**|**HumanEval**|**HumanEval+**|**MBPP**|**MBPP+**|**GSM8K**|**Minerva Math**|**Avg**|**Improvement**|
> |:---:|:---:|:---:|:---:|:---:|:---:|:---:|:---:|:---:|:---:|
> |Train from scratch (600B tokens)|Bidirectional|40.24|38.41|39.40|52.91|55.42|27.63|42.34|-|
> ||Block-wise|48.17|45.73|41.00|58.73|61.33|31.84|47.80|5.46|
> |AR-to-DLM (Qwen3 4B)|Bidirectional|39.02|32.32|39.60|50.00|67.50|39.17|44.60|-|
> ||Block-wise|56.10|51.22|54.60|69.84|82.87|47.02|60.28|15.68|
> |AR-to-DLM (Ministral3 8B)|Bidirectional|50.61|46.95|52.20|70.63|68.46|43.72|55.43|-|
> ||Block-wise|58.54|52.44|53.00|73.81|83.17|55.84|62.80|7.37|

---

> > ### Author Rebuttal · Reviewer_mprB · 2026-04-03
> >
> > Most of my concerns have been resolved, and I maintain my positive rating.

---

### Official Review · Reviewer_tuek · 2026-03-13

**Soundness:** 3
**Presentation:** 4
**Significance:** 3
**Originality:** 3
**Overall Recommendation:** 4
**Confidence:** 3

**Summary:**

This paper studies how to convert pretrained autoregressive LLMs into diffusion language models through continuous pretraining. The core proposal is to replace fully bidirectional conversion with block-wise attention + clean context, and to further reduce the training–test mismatch with position-dependent masking that reflects the model’s left-to-right denoising tendency. Overall, the paper makes a strong empirical case that these design choices lead to substantially better accuracy-throughput trade-offs than prior dLM baselines and competitive trade-offs against strong AR baselines.

**Compliance With Llm Reviewing Policy:**

Affirmed.

**Final Justification:**

The rebuttal adequately addressed my concerns. I maintain my positive assessment.

**Key Questions For Authors:**

- A main claim of the paper is that the gains come from better preserving AR structure during conversion. Could the authors sharpen this causal argument a bit more? For example, it would help to clarify why the current evidence should be interpreted specifically as support for the AR-preservation hypothesis, rather than more generally as evidence that block-wise training is simply a better recipe for this setting.

- The efficiency story is compelling on inference throughput, but the final models also rely on substantial additional continuous pretraining (300B/500B tokens on 128 H100s), and the appendix notes that dLM efficiency advantages diminish at very large batch sizes. I would appreciate a slightly clearer discussion of how the authors want readers to interpret the term “efficient” in light of both the adaptation cost and the serving regime where the gains are strongest.

**Limitations:**

See Weakness and Questions.

**Strengths And Weaknesses:**

### Strengths
- The paper addresses an important and practical problem. The analysis in Sec. 2 is particularly useful: the authors do not just propose block-wise attention, but also show that clean context matters, that token shift is not necessary, and that training/evaluation block sizes have meaningful effects on the final trade-off. This makes the paper more than a single trick, and gives actionable guidance for AR-to-dLM conversion.
- The empirical results are strong and fairly comprehensive. In the main benchmark, Efficient-DLM improves over prior dLMs and shows competitive or better accuracy-throughput trade-offs than AR baselines.
- The paper identifies a clear train/test mismatch in token masking and shows that a moderate positional prior improves accuracy, especially under more aggressive parallel decoding.

### Weaknesses
- The main contribution feels more like a well-executed recipe / systems-design paper than a major conceptual step. The paper is valuable because it organizes several design choices into a strong conversion pipeline, but the individual ingredients themselves are not entirely novel, and the paper’s own framing is often closer to “practical guideline” and “actionable insights” than to a new modeling principle.
- The central explanation that block-wise conversion works because it better preserves pretrained AR behavior is plausible, but still supported mostly by indirect evidence such as weight-drift visualizations and downstream trends. I buy the intuition, but the paper stops a bit short of establishing this mechanism as cleanly as the empirical results themselves.

---

> ### Author Rebuttal · Authors · 2026-03-31
>
> Thank you for your constructive comments! We have answered all your questions as follows:
>
> **Q1. The novelty aspect**
>
> Please refer to our response to Q1 of Reviewer mprB.
>
> ---
>
> **Q2. Key benefits of block-wise attention over bidirectional ones**
>
> This is a great point and will be clarified in the final version. We hypothesize two key benefits of block-wise attention over bidirectional ones: (1) it introduces left-to-right linguistic priors that are more suitable for dLM training; (2) its block-wise causality is closer to AR training, enabling better inheritance of pretrained capabilities.
>
> To disentangle these effects, we train a 4B dLM (same arch as Qwen3 4B) from scratch on 600B tokens using each pattern. The results, along with the 25B AR-to-DLM results on Qwen3 4B (Table 5), are in the table below.
>
> |**Setting**|**Attn Pattern**|**HumanEval**|**HumanEval+**|**MBPP**|**MBPP+**|**GSM8K**|**Minerva Math**|**Avg**|**Improvement**|
> |:---:|:---:|:---:|:---:|:---:|:---:|:---:|:---:|:---:|:---:|
> |Train from scratch|Bidirectional|40.24|38.41|39.40|52.91|55.42|27.63|42.34|-|
> ||Block-wise|48.17|45.73|41.00|58.73|61.33|31.84|47.80|+5.46|
> |AR-to-DLM|Bidirectional|39.02|32.32|39.60|50.00|67.50|39.17|44.60|-|
> ||Block-wise|56.10|51.22|54.60|69.84|82.87|47.02|60.28|+15.68|
>
> From the table, we observe:
>
> (1) **Training from scratch**: Block-wise attention improves average accuracy by +5.46% over the bidirectional one, indicating it is inherently a better pretraining pattern even without AR initialization.
>
> (2) **AR-to-DLM with continuous pretraining**: The gap further increases to 15.68%, showing a much stronger advantage when starting from a pretrained AR model.
>
> These results support both hypotheses, with the larger AR-to-DLM gain highlighting the value of preserving AR distributions for inheriting pretrained capabilities.
>
> ---
>
> **Q3. The adaptation cost of AR-to-DLM**
>
> We highlight two points: (1) Even with a modest budget (25B tokens), AR-to-DLM conversion already achieves strong accuracy (Table 5 of our manuscript); (2) Longer training (e.g., 300B tokens) further improves decoding parallelism by enhancing likelihood estimation and maintaining accuracy under multi-token decoding.
>
> The table below shows AR-to-DLM conversion results on Qwen3 4B with 25B and 300B tokens:
>
> (1) **Decode 1-token per forward:** The 25B model only has a 2.03% gap to the 300B model. This aligns with prior work (*arXiv:2507.11851*), suggesting AR models already possess future-token planning abilities that dLM training can efficiently unlock.
>
> (2) **Decode 3 tokens per forward:** The 300B model preserves accuracy much better, outperforming the 25B model by 8.75%. Notably, 300B tokens correspond to only ~1% of the original 36T-token pretraining cost of Qwen3 4B.
>
> Overall, dLM adaptation can be relatively low-cost while additional training mainly improves decoding parallelism.
>
> |**Eval Setting**|**Training Tokens**|**HumanEval**|**HumanEval+**|**MBPP**|**MBPP+**|**GSM8K**|**Minerva Math**|**Avg**|**Improvement**|
> |:---:|:---:|:---:|:---:|:---:|:---:|:---:|:---:|:---:|:---:|
> |1 tok / forward (threshold=None)|25B|60.37|54.27|59.00|71.43|81.12|45.92|62.02|-|
> ||300B|60.98|56.71|60.00|70.63|86.43|49.54|64.05|+2.03|
> |~3 tok / forward (threshold=0.8)|25B|46.34|41.46|50.40|65.87|80.14|43.08|54.55|-|
> ||300B|60.98|56.10|57.20|69.58|86.88|49.02|63.29|+8.74|
>
> ---
>
> **Q4. The efficiency under larger batch sizes**
>
> dLMs increase computational intensity of AR LMs by co-planning future tokens, leading to a clear trade-off: they improve efficiency in *memory-bound regimes* (e.g., small-batch decoding) by utilizing idle compute, but may reduce efficiency in *compute-bound regimes* (e.g., large-batch serving) due to higher compute cost. Therefore, dLMs are naturally better suited for small-batch scenarios.
>
> That said, two factors help preserve their promise under large batch sizes:
>
> (1) **Hardware aspect:** Compute is scaling much faster than memory bandwidth (e.g., from NVIDIA V100 to NVIDIA B200, ~100× vs. ~10×) in modern GPUs, suggesting future workloads will be increasingly memory-bound, thus favoring dLMs even at scale.
>
> (2) **Algorithm aspect:** Preserving AR capability during AR-to-dLM conversion enables switching between AR (for compute-bound, large-batch scenarios) and dLM (for memory-bound, small-batch scenarios).
>
> For example, in our post-submission experiments on Ministral3 8B with 25B-token continued training, jointly optimizing AR and dLM objectives allows both modes to match or exceed the original AR baseline (see table below), demonstrating the feasibility of a unified model that maintains high efficiency across batch regimes.
>
> |Model|Eval Scheme|HumanEval|HumanEval+|MBPP|MBPP+|GSM8K|Minerva Math|Avg|
> |:---:|:---:|:---:|:---:|:---:|:---:|:---:|:---:|:---:|
> |AR init (Ministral3 8B)|AR|59.15|53.05|68.00|83.60|85.90|69.12|69.80|
> |Ours|dLM|61.59|58.54|64.60|80.42|87.64|65.86|69.77|
> |(Two modes in one model)|AR|62.80|57.93|65.80|82.54|87.79|66.84|70.62|

---

> > ### Author Rebuttal · Reviewer_tuek · 2026-04-03
> >
> > Thank you to the authors for the detailed and thoughtful responses. My concerns have been fully addressed, and I appreciate the clarifications provided. I maintain a positive assessment of the paper.

---

### Decision · Program_Chairs · 2026-04-30

**Decision:**

Accept (regular)

**Comment:**

This paper proposes method for converting pretrained AR LMs to masked diffusion models. The key contributions are a block wise masked attention as well as position based masking ratio. Reviewers are unanimously supportive, acknowledging the soundness of both the methodology and evaluations. The AC agrees with the reviewers and recommend accept.